# Using Word Embeddings to Explore the Learned Representations of Convolutional Neural Networks

## Abstract

As deep neural net architectures minimize loss, they build up information in a hierarchy of learned representations that ultimately serve their final goal. Different architectures tackle this problem in slightly different ways, but all models aim to create representational spaces that accumulate information through the depth of the network. Here we build on previous work that indicated that two very different model classes trained on two very different tasks actually build knowledge representations that have similar underlying representations. Namely, we compare word embeddings from SkipGram (trained to predict co-occurring words) to several CNN architectures (trained for image classification) in order to understand how this accumulation of knowledge behaves in CNNs. We improve upon previous work by including 5 times more ImageNet classes in our experiments, and further expand the scope of the analyses to include a network trained on CIFAR-100. We characterize network behavior in pretrained models, and also during training, misclassification, and adversarial attack. Our work illustrates the power of using one model to explore another, gives new insights for CNN models, and provides a framework for others to perform similar analyses when developing new architectures.

## 1 Introduction

Convolutional Neural Networks (CNNs) manipulate information in the pixels of images by applying a hierarchy of learned functions, and can even outperform humans for some tasks (Karpathy, 2014). While CNNs have become incredibly accurate, they have also become deeper and more complex, making it more difficult to understand why they work, and how they fail. It is not always clear *why* one CNN architecture outperforms another, and when we design networks, the architectural decisions and innovations can be ad hoc, mostly verified by trial and error.

In this paper, we use a technique that allows for the measurement and tracking of information through the layers of a CNN, and offer insights into the function and performance of CNNs. Specifically, we use distributional semantic (DS) models trained on text corpora to explore the hidden representations through the layers of a CNN under a variety of conditions. The methodology is inspired by techniques originally developed to study the brain's semantic representations via neuroimaging data, but has proven useful for understanding CNNs (Dharmaretnam & Fyshe, 2018). Our contributions are:

**1)** Evaluation using **5 times** more concepts than previous work (Dharmaretnam & Fyshe, 2018), including experiments with new architectures (FractalNet), and additional datasets (CIFAR-100).

**2)** An exploration of the behavior of hidden layers during training.

**3)** A case study of the information available in the hidden representations of misclassified images for more complex network architectures.

**4)** An application of the technique to describe hidden representations for adversarial images.

Each of these points illustrate how DS models can help us to understand CNNs, but also how we might use DS models to train better, more robust CNNs. We illustrate here a framework for understanding the behavior of CNNs in a variety of settings to show how this technique could be used to imagine and test better architectures, and possibly protect against adversarial attack.

## 2    RELATED WORK

Distributional models of word meaning (word embeddings) use patterns of word co-occurrence to estimate vector representations for words. These vectors have proven useful for a variety of Natural Language Processing tasks, and have been shown to correlate strongly to human judgments of word similarity (Hill et al., 2015; Bruni & Baroni, 2013), and behavioral norms (Hollis et al., 2017). In fact, several DS models include both text and images to create one joint model (Bruni & Baroni, 2013; Anderson et al., 2013). However, the idea of using DS models to understand CNNs has been largely untouched.

Distributional models have been used in conjunction with CNNs in a variety of ways. CNNs have been trained to predict word vector dimensions as output instead of discrete classification (Frome et al., 2013). Predicting into a space shared with the word vectors allows CNNs to predict for classes not seen during training (zero shot learning) (Socher et al., 2013; Lazaridou et al., 2014). Previously, distributional semantic models have also been used to explore CNNs (Dharmaretnam & Fyshe, 2018). Compared to Dharmaretnam & Fyshe (2018), our work includes 5 times more concepts, new architectures (FractalNet), additional datasets (CIFAR-100), and additional explorations of the behavior of the hidden layers during training, and for adversarial examples.

The interpretation of CNNs has taken many forms, but most have relied on visual exploration of the images that most "excite" a neuron (Yosinski et al., 2015), or the areas of an image that most contribute to a prediction (Zeiler & Fergus, 2014). Semantic parts (e.g. wheels, legs) have also been identified within CNN representations (Gonzalez-Garcia et al., 2018), another piece of evidence that CNNs capture semantic meaning. There have also been several variants of Canonical Correlation Analysis (CCA) proposed to project the hidden layers into a shared representational space with either the raw image pixels or the layers of another CNN (Morcos et al., 2018; Saini & Papalexakis, 2018; Raghu et al., 2017). In this work, we use an independent model trained on very different data (text) as a sort of "third-party" evaluation of the information that exists in the layers of the CNN. This allows us to move beyond the similarities that exist image space (e.g. many animals are pictured in outdoor scenes) and instead correlate to another notion of semantics built from the usage of the word associated with the concept.

## 3    METHODOLOGY

Our initial experiments reproduce and expand upon a previous study of the hidden representations of two popular CNNs both trained on ImageNet (Deng et al., 2009): ResNet-50, and Inception-v3. We proceed with experiments on FractalNet, which we train on CIFAR-100 (Krizhevsky & Hinton, 2009), to study both the behavior of a network trained on a different image classification dataset, and the behavior of a network during the training process.

**ResNet-50:**    The ResNet architecture (He et al., 2015) was introduced as an entry to the 2015 Large Scale Visual Recognition Challenge (Russakovsky et al., 2015; Szegedy et al., 2015a). ResNet introduces residual blocks, which contain residual connections that give each block's final layer access to the block's original input. This helps both with stability during training, and also with predictive accuracy. We study ResNet-50, the 50 layer variant that achieved a top-5 error rate of 5.25% on the ILSVRC 2015 test dataset. We studied the 49 activation layers spread across 16 residual blocks in this network.

**Inception-v3:**    Inception-v3 is a variant of the original GoogLeNet architecture (Szegedy et al., 2015b). It has 94 convolutional blocks followed by ReLu activations. The network has nine inception modules. The outputs of each branch within an inception module are concatenated before they are passed to the next inception layer. We studied both activation and filter concatenation layers ( *mixed* layers in *Keras* (Chollet et al., 2015)).

**FractalNet:**    FractalNet is an interesting architecture trained with both deep and shallow connection paths made up of fractal expansions of the same base architecture (Larsson et al., 2016). Unlike ResNet (a network with comparable deep and shallow connection paths), the FractalNet architecture continues to improve with added depth (though with diminishing returns).

### 3.1 DISTRIBUTIONAL SEMANTIC MODELS

We selected SkipGram as our Distributional Semantic (DS) model for studying the semantic representations in CNNs. Previous work showed SkipGram's performance to be approximately equal to other DS models (Dharmaretnam & Fyshe, 2018), and its coverage over ImageNet classes is higher. SkipGram is part of the Word2vec package, and is trained on the Google News dataset to predict context words given a central word (Mikolov et al., 2013). We used the 300-dimensional version of the model.

### 3.2 CONCEPT SELECTION

Two of the CNNs used in our study are pretrained on ImageNet (Deng et al., 2009) which has 1000 labeled image classes organized to align with the WordNet hierarchy (Fellbaum, 1998). ImageNet classes often are a list of synonymous concepts (e.g. class 286: "cougar, puma, catamount, mountain lion, painter, panther, Felis concolor"). For these cases, we use the vector of the first word that matches a SkipGram vector, and we were able to find matches to 838 ImageNet classes. Matching word vectors to CIFAR-100 was more straightforward, as most of the classes are single words (or could be represented as a single word), and all 100 were present in the SkipGram word list.

### 3.3 ANALYZING LEARNED REPRESENTATIONS

Following Dharmaretnam & Fyshe (2018), we randomly selected 5 distinct images for each of the $w = 838$ matched concepts from the ImageNet cross-validation dataset (Russakovsky et al., 2015). All images were rescaled to $224 \times 224$ for ResNet-50 and VGG-16, and $299 \times 299$ for Inception-v3. After resizing, the pixel values were mean normalized, and we generated representations for *each layer*[1] of each CNN. This results in 5 matrices $I \in \mathbb{R}^{w*k}$ where $k$ is the dimension of a flattened CNN layer. Thus, for a network with 50 layers, there will be 5 $I$ matrices per layer, for a total of 250 matrices. Each row in a matrix $I$ represents the hidden representation of one image example of a concept extracted from one layer of the CNN.

We then compute the Pearson correlation of every concept in a CNN's matrix $I$ with every other concept in the same $I$ matrix, resulting in a correlation matrix $C_I \in \mathbb{R}^{w*w}$. Thus, every row $C_I(i)$ in the correlation matrix $C_I$ represents the similarity of the hidden representation of a concept $i$ with every concept $j = 1 \rightarrow w$ in the matrix $I$. This process is repeated for all the 5 CNN matrices $I$ resulting in 5 CNN correlation matrices $C_I$ per layer.

We extract the SkipGram vectors for the same 838 concepts, resulting in a matrix $D \in \mathbb{R}^{w*n}$, where $w$ is the number of concepts and $n$ is the number of dimension of the SkipGram vectors. We then compute the Pearson correlation of every word in word vector matrix $D$ with every other word in the matrix resulting in the correlation matrix $C_D$ where $C_D \in \mathbb{R}^{w*w}$. Row $i$ of the matrix $C_D(i)$ represents the similarity of a word $i$ with every word $j = 1 \rightarrow w$ in the matrix $D$. Now we have matrices $C_I$ and $C_D$ which represent the similarity of concepts in CNN and DS space.

How can we compare the similarities between representations in $C_I$ and $C_D$? We could just compute the correlation of the upper triangle of $C_I$ and $C_D$, but this would obfuscate which concepts are best represented in each layer. Instead, we use the 2 vs. 2 test (Dharmaretnam & Fyshe, 2018) that allows us to better explore the representations at the level of individual concepts. For the 2 vs. 2 test, we select the rows corresponding to two concepts ($c_1$ and $c_2$) from our correlation matrices $C_I$ and $C_D$. We then omit the columns corresponding to the same two concepts ($c_1$ and $c_2$), resulting in vectors with $w - 2$ elements. These vectors represent the correlation of the representations of $c_1$ and $c_2$ to every other concept in both CNN and DS space. Let's rename the reduced vectors as $c_{I_1}, c_{I_2}$ from the CNN correlation matrix $C_I$, and $c_{D_1}, c_{D_2}$ from the DS correlation matrix $C_D$. In a 2 vs. 2 test, the correlation of the concepts are compared to test if the correlation of the correctly matched pairs:

$$\text{corr}(c_{I_1}, c_{D_1}) + \text{corr}(c_{I_2}, c_{D_2}) \tag{1}$$

is greater than the correlation of the mismatched pairs:

$$\text{corr}(c_{I_1}, c_{D_2}) + \text{corr}(c_{I_2}, c_{D_1}) \tag{2}$$

A 2 vs. 2 test is considered to pass if Eq. 1 is greater than Eq. 2. The test is repeated for all possible pairs of concepts in our set of 838 concepts. This results in $\binom{w}{2}$ tests for a dataset with $w = 838$

---

[1]A layer can be any node in the computation graph. Here, we focus on activation and concatenation layers.

concepts. The 2 vs. 2 accuracy is the percentage of 2 vs. 2 tests passed, and chance is 50%. Note that this method is based entirely on correlation, and so can only detect linear relationships. Still we were impressed with the results this simple approach yielded. Future work might consider learning a (possibly non-linear and/or regularized) mapping to further explore the relationship between representational spaces.

The 2 vs. 2 tests were repeated for the 5 $C_I$ matrices independently, and the scores were averaged to get a single score for a given layer of the CNN. This accounts for variability across images for a single concept in ImageNet (though in practice we found the average variation across the 5 matrices to be very small, e.g. 0.0064 for Inception-v3). The whole process is then repeated for each layer in each CNN.

## 4 STUDYING MISCLASSIFICATION AND ADVERSARIAL EXAMPLES

We were also interested in studying when and how CNNs fail. For this, we explore unaltered images that are misclassified by a CNN, and adversarial images designed to mislead CNNs. We search through the hierarchical layers of the CNN to identify layers where misclassifications emerge, illustrating the vulnerabilities in CNN architectures, and providing a road map to debug and improve CNNs.

Out of the 838 total Skip-Gram concepts, we collected a set of 89 images which were incorrectly classified by ResNet-50, and 61 incorrectly classified by Inception-v3.

To study adversarial examples, we randomly selected 100 source concepts. For each source concept, we select 6 target concepts for a total of 600 adversarial targets. To select targets, we calculate the correlation of the source concept's word vector with every other concept's word vector. We select six target concepts including the most similar target concept, the least similar target concept, and four other target concepts spaced evenly between. For each source and target pair, we perform a targeted adversarial attack against Inception-v3 using v2.0.0 of Cleverhans (Papernot et al., 2017). Our attack algorithm is the Momentum Iterative Method with an $L_\infty$ norm perturbation bound of $\epsilon = 0.3$ using a decay factor of $\mu = 1.0$ over 20 iterations (Dong et al., 2017). We then tested each of the 600 adversarial images and discarded the four images that were not predicted to be the target class (failed attacks), resulting in 596 total adversarial images. Examples of the adversarial images are available in Appendix A.

For a given misclassified concept $i$ (either an unaltered misclassified image or an adversarial image), there is a true class $t_i$ and a predicted class $p_i$. We need to compare these true and predicted classes to the representations of other concepts, and to ensure accuracy we select exclusively from the set of **correctly predicted** concepts. We randomly sampled 100 correctly predicted concepts, and extracted hidden layer representation using a single image per concept. This resulted in a matrix $I_{\text{correct}} \in \mathbb{R}^{100*k}$ where $k$ is the dimension of the flattened CNN layer. Similarly, word vectors corresponding to the same 100 concepts were extracted from the Skip-Gram model. Lets call this matrix $D_{\text{correct}} \in \mathbb{R}^{100*n}$ where $n$ is the dimension of the Skip-Gram word vectors. The 100 concepts represented in $I_{\text{correct}}$ and $D_{\text{correct}}$ never include $t_i$ or $p_i$.

Next, for each misclassified concept $i$, hidden representations were extracted for each CNN layer, and correlations were computed with every concept in $I_{\text{correct}}$ resulting in the vector $i_{\text{misclassified}} \in \mathbb{R}^{100}$. The word vectors corresponding to true class $t_i$ and predicted class $p_i$ were also extracted, and correlations computed with every concept in $D_{\text{correct}}$ resulting in two vectors $d_{\text{true}}$ and $d_{\text{predicted}}$, both of dimension $\mathbb{R}^{100}$. The vector $i_{\text{misclassified}}$ represents the correlation of concepts in CNN vector space whereas $d_{\text{true}}$ and $d_{\text{predicted}}$ represents correlation of concepts in word vector space. We then check to see if the $i_{\text{misclassified}}$ is more correlated to $d_{\text{true}}$ or $d_{\text{predicted}}$:

$$\text{corr}(i_{\text{misclassified}}, d_{\text{true}}) \overset{?}{>} \text{corr}(i_{\text{misclassified}}, d_{\text{predicted}}) \tag{3}$$

This is the 1 vs. 2 test (Dharmaretnam & Fyshe, 2018), and the chance accuracy is again 50%. The test is repeated for all misclassified concepts and adversarial images, which gives us a measure of if the semantic information of the true class exists anywhere in the CNN's hierarchy.

### 4.1 TESTING FOR STATISTICAL SIGNIFICANCE

The 2 vs. 2 and 1 vs. 2 tests were designed to study the relationship between concepts in different vector spaces. The chance accuracy for both these tests is 50%, but we need to calculate a confidence

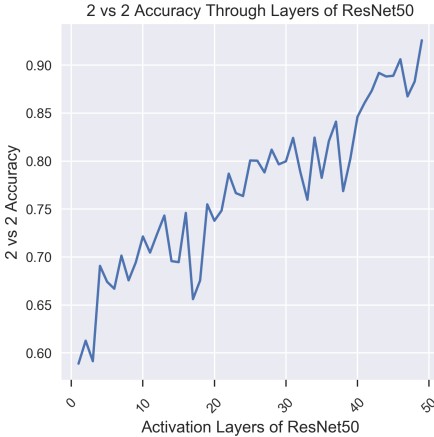
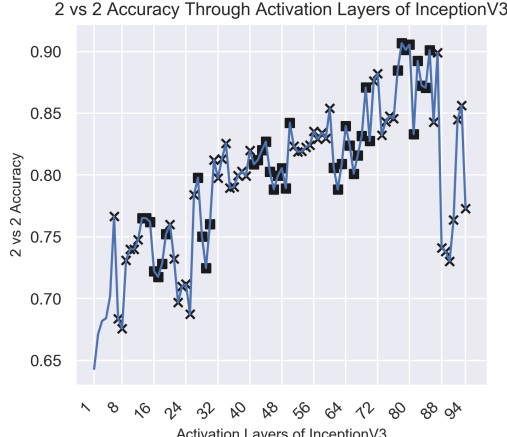

Figure 1: 2 vs. 2 accuracy for the activation layers of ResNet-50.

Figure 2: 2 vs. 2 accuracy for the layers of Inception-v3. Inception modules introduce parallel paths each containing activation nodes. To demarcate Inception module boundaries, marker types alternate.

interval around 50%, outside of which results are significant. We do this by running 1000 permutation tests (Wasserman, 2004), approximating p-values for our observed 2 vs. 2 and 1 vs. 2 accuracies, and correcting for multiple comparisons using Benjamini-Hochberg-Yekutieli (BHY) false discovery rate correction (Benjamini & Yekutieli, 2001). This is standard practice for estimating significance, while making very few statistical assumptions.

## 5    RESULTS AND DISCUSSION

**CNNs Learn Semantics from Images**    Figures 1 and 2 show a general upward trend in 2 vs. 2 accuracy through the layers of both the convolutional networks (see also the annotated architecture diagrams in the Appendix B and C). However, we also observed that the accuracy increase through the layers of ResNet-50 and Inception-v3 was very noisy. In Figure 1, we observed that the 2 vs. 2 accuracy in the layer immediately before or after a residual block is always higher than the layers inside a residual block. For example, this can be seen in layer 19 of Figure 1 (the end of a residual block) when the 2 vs. 2 accuracy vastly improves. This may be explained by residual learning theory. The ResNet-50 architecture is composed of *residual blocks* which contain convolutional layers internally and a skip-connection that connects the input of the residual block to the final layer (He et al., 2015). Conceptually, each residual block is a module that calculates a small change $F_{(x)}$ for a given input $x$ to the residual block. The *add* layer at the end of residual blocks combines $F_{(x)}$ with original input $x$ via the skip connections. Combining $F_{(x)}$ with $x$ provides additional information to the activation layer after a residual block. This effect is also very clear in the annotated ResNet-50 architecture diagram (Appendix Figure 8). These results provide a semantic argument for the effectiveness of residual learning theory, and illustrate the power of the 2 vs. 2 technique

Figure 2 shows the 2 vs. 2 accuracy of activation layers of Inception-v3. The Inception-v3 architecture consists of three different types of inception modules occurring in series (Szegedy et al., 2015b). Within each module, multiple convolutional and pooling operations happen in parallel that are concatenated at the end of the module. Because of these parallel connections, the activations along the x axis in Figure 2 cannot be linearly ordered. For this reason, the points are marked with alternating markers, which indicate the module boundaries (i.e. all activations within a module appear in a block with the same marker type). Similar to ResNet-50, within an inception module we see 2 vs. 2 accuracy decreases and increase again at the end of the module when mixing the various parallel convolution operations together (this is most clear in the architecture diagram, Appendix Figure 9). Interestingly, shallower parallel connections seem to maintain 2 vs. 2 accuracy better than paths with many convolutions, which implies that the lower-dimensional convolutions may be driving some of the early performance. It should also be noted that the average pooling at the end of the last inception module also provides a tremendous boost in network performance while reducing the number of

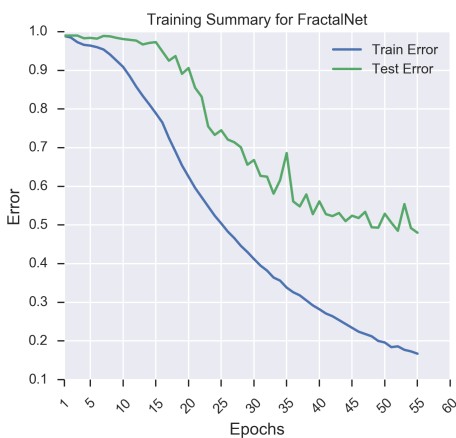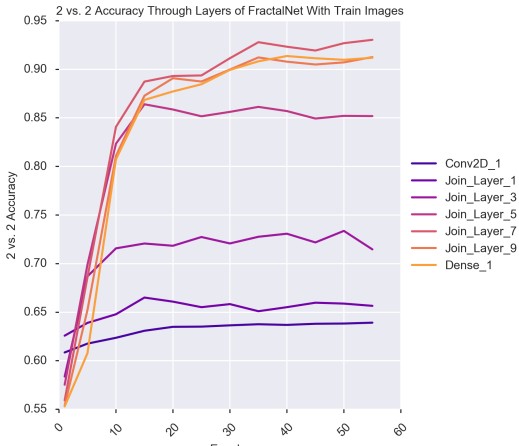

Figure 3: The emergence of semantic information during training of FractalNet on CIFAR-100. Left: training and test error for the first 60 epochs training. Right: the 2 vs. 2 accuracy for layers of FractalNet during training. We show 2 vs. 2 accuracy at the first convolutional layer, the end of every second fractal network block, and the last fully connected layer. The initial layers of the CNN learn semantics before the later layers, but later layers continue to improve in later epochs.

parameters. These two case studies show how using this framework can allow those building new architectures to quantitatively test which innovations are truly benefiting the performance of a new system.

## 5.1 TRAINING CNNs

The study of the development of semantic information in the hidden layers of CNNs during training could offer additional insights. We trained *FractalNet* (Larsson et al., 2016) on CIFAR-100 (Krizhevsky & Hinton, 2009), using methods described in the original paper. At every five epochs during the training process, we extracted the hidden layers of the network and performed 2 vs. 2 tests using the images from the training set of CIFAR-100. The results for the first 60 epochs appears in Figure 3, and Appendix D shows all 400 epochs. Note that the initial layers of the CNN learn semantics before the later layers, and that they learn semantics within the first few epochs of training. The 2 vs. 2 accuracy for the initial layers remains constant throughout the remainder of training, and the 2 vs. 2 accuracy for the later layers continues to increase until the end of the first 60 epochs. Thus, a significant part of later learning is driven by the middle and later layers of the CNN. We also measured the 2 vs. 2 accuracy for test images, and found it to be 2%-3% lower than the accuracy on train images.

We also noted that as the network starts to overfit, the 2 vs. 2 accuracy curve becomes noisy, raising the question of what the 2 vs. 2 accuracy would look under the regime of permuted labels (Zhang et al., 2016). We trained *FractalNet* after randomly permuting both the train and test image labels, and trained the until we achieved 99.9% training accuracy. As expected, the test accuracy was close to chance (1%). We conducted the 2 vs. 2 tests for various layers of CNN. We found that, even though the network achieves close to perfect classification accuracy on train images, the 2 vs. 2 accuracy stays consistently low. This points to another method for identifying overfitting during CNN training: *a network fitting to noise does not learn semantics*.

## 5.2 MISCLASSIFICATIONS IN CNNs

No CNN is perfect, so every CNN misclassified some images. This prompted us to search through the learned representations to see if the information required to make the correct prediction exists in any layer of ResNet-50 or Inception-v3. Contrary to previous work by Dharmaretnam & Fyshe (2018), we found no points to be statistically above chance once we corrected for multiple comparisons. Because the results were very noisy, we also tried averaging 2 vs. 2 accuracy for activation nodes within residual blocks for ResNet-50, and within Inception modules for Inception-v3. Figures 4

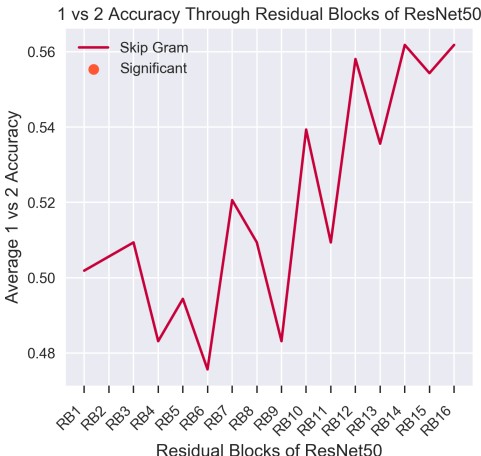

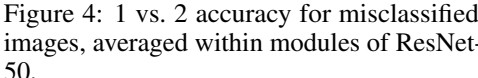

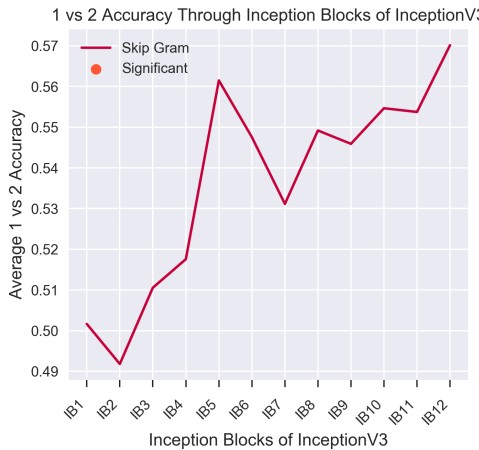

Figure 4: 1 vs. 2 accuracy for misclassified images, averaged within modules of ResNet-50.

Figure 5: 1 vs. 2 accuracy for misclassified images, averaged within modules of Inception-v3.

and 5 show the block/module average 1 vs. 2 accuracy for ResNet-50 and Inception-v3 respectively. Even with this averaging scheme, we found no points above chance. So, though a less sophisticated network (VGG-16) was found to represent some signatures of the true class (reaching nearly 0.65 1 vs. 2 accuracy), the more modern ResNet-50 and Inception-v3 have much lower 1 vs. 2 accuracy and do not show this characteristic.

To explain these results, we surveyed the mistakes made by these two networks by calculating the cosine similarity between the word vectors corresponding to the true and predicted class. We compared this against 89 mistakes made by VGG-16, for which misclassifications had been previously studied (Dharmaretnam & Fyshe, 2018). We found that VGG-16 makes more egregious classification mistakes (average of 0.22 cosine similarity) as compared to Inception-v3 and ResNet-50 (0.27 and 0.29 cosine similarity, respectively). When there is a high similarity between true and predicted concepts, it becomes difficult for the 1 vs. 2 tests to separate the semantics of the true and predicted class. Conversely, less similar true and predicted classes make it easier to distinguish the semantics of true class from the predicted class. This helps to explain why VGG-16 has statistically significant 1 vs. 2 accuracy in previous work, whereas ResNet-50 and Inception-v3, which make more acceptable misclassifications, do not.

## 5.3 ADVERSARIAL EXAMPLES

Adversarial examples are one of the more serious threats to the adoption of CNNs for practical use, and the vulnerability of neural systems to relatively minor perturbations is a growing concern. We explored the hidden representations through the layers of Inception-v3 for adversarial examples to better understand the internal representation of CNNs during adversarial attack. We focus on the Inception-v3 model here as it has the highest ImageNet challenge top-1 accuracy of the three models (Canziani et al., 2016; Russakovsky et al., 2015). Like the study of misclassification, we use the 1 vs. 2 test, which passes if the true class ($d_{true}$) is closer than the adversarial class ($d_{predicted}$) to the CNN correlation vector ($i_{misclassified}$).

Figure 6 shows the 1 vs. 2 accuracy through layers of Inception-v3 for the 596 adversarial examples, broken down into six target concepts based on the level of word vector correlation between the target and source concept. For simplicity here, we show only the 1 vs. 2 accuracy for "Mixed" layers, which join the result of several parallel paths within a module. On average, in earlier layers of the network, the 1 vs. 2 accuracy is higher, but by the later layers of the network, the hidden representations have become more similar to the adversarial class ($d_{predicted}$), pushing the 1 vs. 2 accuracy below 50. Below 50, the hidden representations are correlated to the adversarial concept.

Compared to other target concepts, the most similar target concept has lower correlation in the earlier layers and higher correlation in the later layers. This is because the semantic representation for two highly related concepts is more difficult to disambiguate, so the 1 vs. 2 accuracy will be closer to

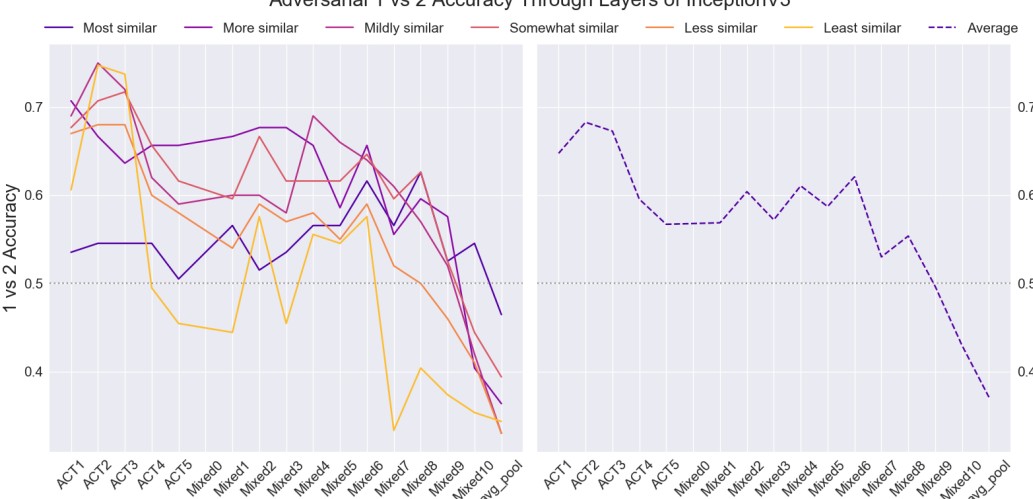

Figure 6: 1 vs. 2 accuracy through layers of Inception-v3 for adversarial examples. When the accuracy is above 50, there is more evidence for the true class than the adversarial class. Left: Average 1 vs. 2 accuracy for the six categories of adversarial targets. Right: average 1 vs. 2 accuracy for all six categories of adversarial targets. ACT: activation layer, Mixed: concatenation block at the end of a module.

chance. For all other target concepts, which are less related to the source concept, we see a pattern much more similar to the average: high 1 vs. 2 accuracy in early layers, and lower 1 vs. 2 accuracy for later layers. This implies it is the later layers that are being targeted by adversarial attacks, probably by an accumulation of small perturbations through the network.

These results suggest interesting avenues for further exploration. The results in Figures 1 and 2 show that, for non-adversarial examples, we should expect there to be a correlation between the predicted word vector's semantic representation and the convolutional neural network's semantic representation. This correlation should increase and become high in the later layers. Because adversarial attacks work through minor perturbations which leverage the details of the network's decision boundary, they may not display this pattern. The detection of this behavior could be a potential mechanism for defense against the current generation of adversarial attacks. One potential implementation would calculate a 1 vs. 2 test at each layer using the input image, the predicted label, and one other label. This would be repeated for all labels. The resulting average 1 vs. 2 signature may be an effective indicator for verifying the semantic representation pattern is idiomatic of a non-adversarial input.

Conversely, this also suggests a mechanism for improving adversarial attacks, which currently suffer from transferability problems, possibly because decision boundaries can vary widely between networks (Liu et al., 2017). Since all three neural networks we studied are learning representations that produce high 2 vs. 2 results against Skip-Gram, it may be possible to implement an adversarial attack which is regularized toward producing hidden representations that have high correlation to that of the *target* class rather than purely optimized for a single network's decision boundary. This sort of attack could be more robust to new and different networks.

## 6    SUMMARY

Here, we studied the representations learned by CNNs, using the representations learned by SkipGram. We measured the behavior of fully trained networks, a network during training, and during the processing of a misclassified or adversarial images. Our results point to several new avenues for training CNNs, and also for characterizing their behavior during failure modes. This is a new approach to understanding CNNs that brings quantifiable interpretability without requiring the visual inspection of images or activation patterns. This method of analysis could further operationalize the development of deep learning architectures by providing a framework within which to reason about changes to an architecture, and what each new architectural innovation brings in our quest to help computers understand our world.

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

APPENDIX

## A    GENERATED ADVERSARIAL EXAMPLES

We generated a series of targeted adversarial examples against InceptionV3 using a recent adversarial attack. In Figure 7 we show eight example images from the attack with our chosen parameters. While the pixel changes to the image are visually detectable, they are minor and the true semantic concepts of the images are clearly retained.

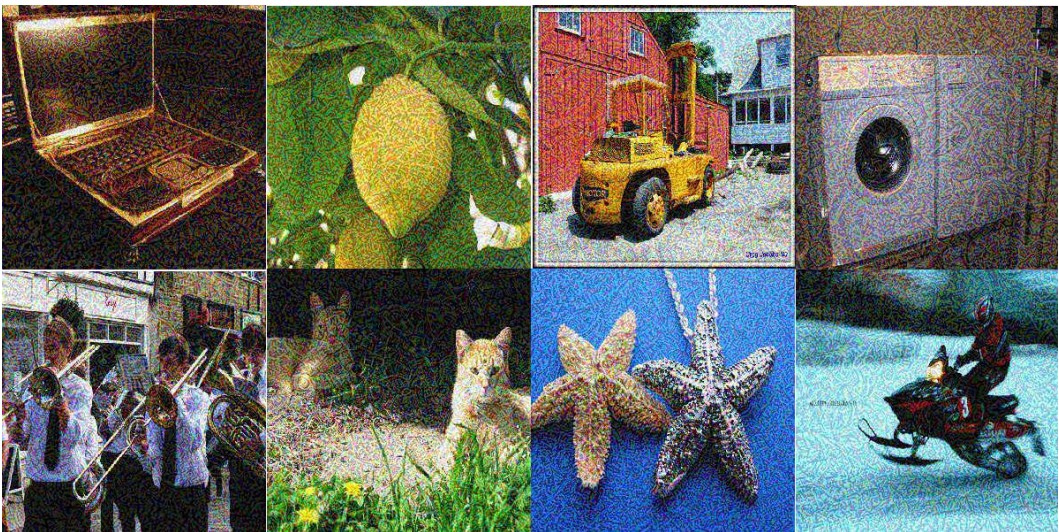

Figure 7: A sample of eight targeted adversarial examples.

## B    ANNOTATED RESNET ARCHITECTURE DIAGRAM

We created an network architecture diagram graphic of ResNet-50 which is shown in Figure 8. We annotated the activations on the architecture diagram using a color gradient correlating to the 2 vs 2 accuracy for that layer. Layers which are more strongly represent the semantics of the true label score higher on the 2 vs 2 test. We can see here that the semantic representation becomes stronger as the ResNet network becomes deeper.

However, a unique feature of interest is the skip connections in ResNet. A consistent pattern through the layers augmented with skip connections is that the semantic representation will decrease through the branch, until the final *add* layer with the skip connection which leads to a higher semantic representation than before the branch. This provides evidence that skip connections are useful techniques for passing along semantic representation in deeper networks.

## C    ANNOTATED INCEPTIONV3 ARCHITECTURE DIAGRAM

We created an network architecture diagram graphic of Inception-v3 which is shown in Figure 9. We annotated the activations on the architecture diagram using a color gradient correlating to the 2 vs 2 accuracy for that layer. Layers which are more strongly represent the semantics of the true label score higher on the 2 vs 2 test. As with the prior two networks, we continue to see a general pattern of increasing semantic representation through later layers of the network.

Inception-v3 has a complicated architecture, which makes it a good candidate for exploration using this semantic annotation method. We see a similar pattern to ResNet-50, where parallel layers within Inception-v3 blocks may decrease in semantic representation before increasing again at the mixing layer of the block. However, not all parallel layers decreasing in semantic representation suggesting some parallel layers may provide more semantic value than others within the block.

Of particular note in this network is the very large decrease in semantic representation in the final layers for some parallel components of the block. This annotation method may be a useful tool for

identifying steps in the network which are not improving semantic representation, and removal or adjustment or those layers may provide a positive affect on classification accuracy.

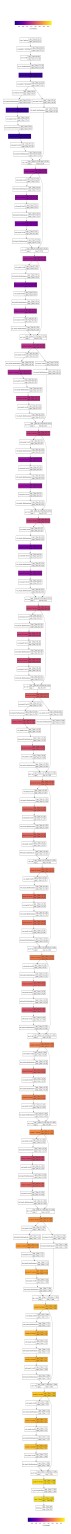

Figure 8: The 2 vs. 2 accuracy through architecture diagram of ResNet-50. The architecture diagram of ResNet-50 is annotated with 2 vs. 2 accuracy of layers against Skip-Gram word-vectors (He et al., 2015). This is a high resolution image and can be zoomed/viewed in a pdf.

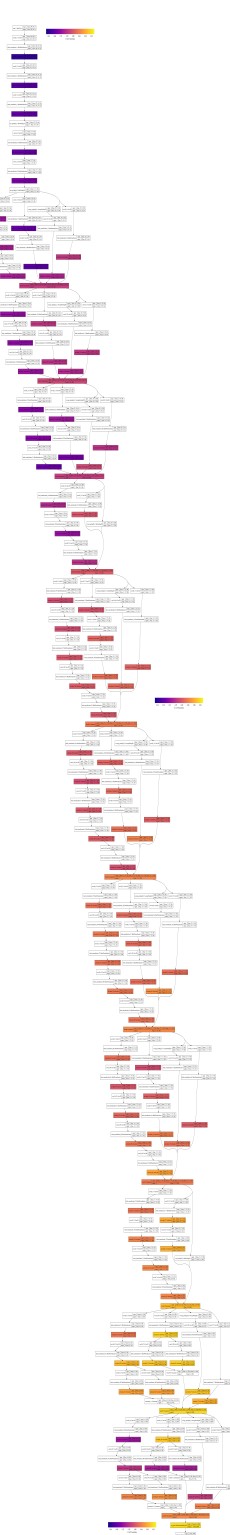

Figure 9: 2 vs. 2 accuracy through architecture diagram of Inception-v3. The architecture diagram of Inception-v3 is annotated with 2 vs. 2 accuracy of layers against Skip-Gram word-vectors (Szegedy et al., 2015b). This is a high resolution image and can be zoomed/viewed in a pdf.

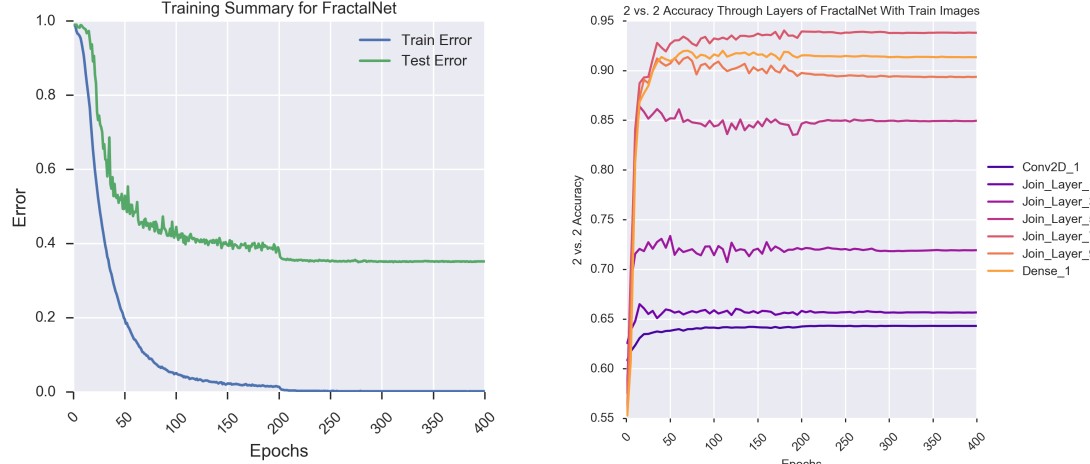

Figure 10: The emergence of semantic information during training of FractalNet on CIFAR-100. Left: training and test error for the 400 epochs of training. Right: the 2 vs. 2 accuracies for layers of FractalNet during training.

## D FRACTALNET 2 VS. 2 ACCURACY FOR 400 EPOCHS

The train/test error and 2 vs. 2 accuracy for the entire 400 epochs of training of the FractalNet is shown in the Figure 10. The change in behavior around 200 epochs corresponds to a reduction in learning rate.

