# OpenReview forum: "Using Word Embeddings to Explore the Learned Representations of Convolutional Neural Networks"
_ICLR.cc/2019/Conference_

### Official Review · AnonReviewer1 · 2018-10-29
**Limited contribution. In addition, it's hard to identify the contribution of this paper.**

**Rating:** 4
**Confidence:** 2

**Review:**

This paper extends the previous work (Dharmaretnam & Fyshe, 2018), which provided a analytic tool for understanding CNNs through word embeddings of class labels. By analyzing correlations between each CNN layers and class labels, enables to investigates how each layer of CNNs work, how much it performed well, or how to improve the performance.

I felt it is little hard to read this paper. Although the short summary of contributions of this paper in the Introduction, I could not easily distinguish contributions of this paper from the ones of the previous work. It's better to explicitly explain which part is the contributions of this paper in detail. For example, "additional explorations of the behavior of the hidden layers during training" is not clear to me because this expression only explain what this paper do briefly, not what this paper is actually different from the previous work, and how this difference is important and crucial.

Similarly, I could not understand why adding concepts, architectures (FractalNet), datasets (CIFAR-100) is so important. Although this paper states these changes are one of the contributions, it is unclear whether these changes lead to significant insights and findings which the previous work could not find, and whether these findings are so important as contributions of this paper. Again, I think it is better to describe what main contributions of this paper are in more detail.

---

### Official Review · AnonReviewer2 · 2018-11-02
**Too incremental**

**Rating:** 4
**Confidence:** 4

**Review:**

The authors apply an existing method (mainly 2 vs 2 test) to explore the representations learned by CNNs both during/after training.

## Strength

The analysis of misclassification and adversarial examples is interesting. The authors also propose potential ways of improving the robustness of DNNs for adversarial examples.


## Weakness
1. It seems to me that the methodological novelty is limited, which mainly follows [The Emergence of Semantics in Neural Network Representations of Visual Information](http://aclweb.org/anthology/N18-2122). For example this paper extensively applies 2 vs. 2 test which was established in previous works.
Furthermore, the first claimed contribution of 5 times more concepts than previous work does not result in any significant difference from the previous approaches.

2. The analysis presented in this work does not really give new insights. For example, isn’t “a network fitting to noise does not learn semantics” obvious to the community?

Some of the subsection titles are misleading. For example in Section 5, the claim of “CNNs Learn Semantics from Images” is mainly proposed in a previous work, but the way of presentation sounds like this is a contribution of this work.

---

### Official Review · AnonReviewer3 · 2018-11-05
**Rough idea. The proposed relationship is not properly confirmed.**

**Rating:** 3
**Confidence:** 4

**Review:**

The authors propose a new method of measuring a knowledge within the learned CNN: the representations of CNN layers and word2vec embeddings and compared, and the similarity between them are calculate. The authors claim that the similarity score increases with learning time, and the higher layers of CNN have more similarity to word2vec embeddings than the lower layers..

CNN and word2vec use different datasets. CNN uses the vision pixels and word2vec uses the words in the sentences. A certain amount of representation patterns can be expected to be shared, but surely the extent is limited (correlation 0.9 in Fig. 1). Because of this limitation, the proposed similarity measure must not be claimed as the measure of knowledge accumulation in CNN.

In addition, the authors have to be precise in defining the measure and provide the information captured by the measure. In the literature, I can see “something” is shared by the two algorithms but do not know what is this “something.” The authors claim that “semantics” are shared, but replacing “semantics” to “something” does not make any difference in this manuscript. Further investigations and confirmations are needed to report which information is actually similar to each other.

Minor: the 1 vs. 2 accuracy measure is not defined.

In summary, the proposed measure may capture some information but the explanation about this information is unclear. The information seems to be a rough similar pattern of concept representations. Further rigorous investigation of the proposed measure is necessary to confirm which information is captured. The current version is not sufficient for acceptance.

---

### Meta-Review · Area_Chair1 · 2018-12-14
**weak novelty and major clarity issues**

**Confidence:** 5
**Recommendation:** Reject

**Metareview:**

The paper aims to study what is learned in the word representations by comparing SkipGram embeddings trained from a text corpus and CNNs trained from ImageNet.

Pros:
The paper tries to be comprehensive, including analysis of text representations and image representations, and the cases of misclassification and adversarial examples.

Cons:
The clarity of the paper is a major concern, as noted by all reviwers, and the authors did not come back with rebuttal to address reviewers' quetions. Also, as R1 and R2 pointed out the novelty over recent relevant papers such as (Dharmaretnam & Fyshe, 2018) is not clear.

Verdict:
Reject due to weak novelty and major clarity issues.